# Multimodal Trajectory Prediction Conditioned on Lane-Graph Traversals

**Nachiket Deo**[1*]      **Eric M. Wolff**[2]      **Oscar Beijbom**[2]
[1]UC San Diego      [2]Motional
ndeo@ucsd.edu, {eric.wolff, oscar.beijbom}@motional.com

**Abstract:**

Accurately predicting the future motion of surrounding vehicles requires reasoning about the inherent uncertainty in driving behavior. This uncertainty can be loosely decoupled into lateral (e.g., keeping lane, turning) and longitudinal (e.g., accelerating, braking). We present a novel method that combines learned discrete policy rollouts with a focused decoder on subsets of the lane graph. The policy rollouts explore different goals given current observations, ensuring that the model captures lateral variability. Longitudinal variability is captured by our latent variable model decoder that is conditioned on various subsets of the lane graph. Our model achieves state-of-the-art performance on the nuScenes motion prediction dataset, and qualitatively demonstrates excellent scene compliance. Detailed ablations highlight the importance of the policy rollouts and the decoder architecture.

**Keywords:** Motion prediction, autonomous vehicles, graph neural networks

## 1   Introduction

To safely and efficiently navigate through complex traffic scenes, autonomous vehicles need the ability to predict the intent and future trajectories of surrounding vehicles. There is inherent uncertainty in predicting the future, making trajectory prediction a challenging problem. However, there's structure to vehicle motion that can be exploited. Drivers usually tend to follow traffic rules and follow the direction ascribed to their lanes. High definition (HD) maps of driving scenes provide a succinct representation of the road topology and traffic rules, and have thus been a critical component of recent trajectory prediction models as well as public autonomous driving datasets.

Early work [1] encodes HD maps using a rasterized bird's eye view image and convolutional layers. While this approach exploits the expressive power of modern CNN architectures, rasterization of the map can be computationally inefficient, erase information due to occlusions, and require large receptive fields to aggregate context. The recently proposed VectorNet [2] and LaneGCN [3] models directly encode structured HD maps, representing lane polylines as nodes of a graph. VectorNet aggregates context using attention [4], while LaneGCN proposes a dilated variant of graph convolution [5] to aggregate context along lanes. These approaches achieve state-of-the-art performance using fewer parameters than rasterization-based approaches.

The above methods represent the HD map as a graph and encode the input context into a single context vector as shown in Fig.1. The context vector is then used by a multimodal prediction header [1, 6] to output multiple plausible future trajectories. The prediction header thus needs to learn a complex mapping, from the entire scene context to multiple future trajectories, often leading to predictions that go off the road or violate traffic rules. In particular, the prediction header needs to account for both *lateral* or *route* variability (e.g. will the driver change lane, will they turn right etc.) as well as *longitudinal* variability (e.g. will the driver accelerate, brake, maintain speed). This decoupling of routes and motion profiles for trajectories has been used in path planning [7, 8], and more recently in prediction [9].

---

*Work done during an internship at Motional.

5th Conference on Robot Learning (CoRL 2021), London, UK.

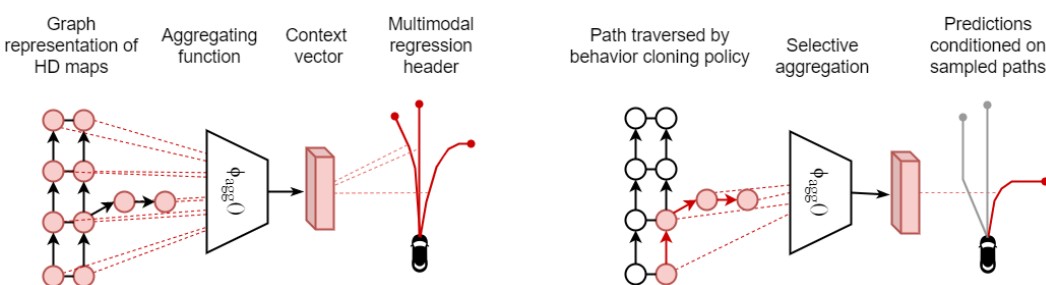

Figure 1: **Overview of our approach.** We encode HD maps and agent tracks using a graph representation of the scene. However, instead of aggregating the entire scene context into a single vector and learning a one-to-many mapping to multiple trajectories, we condition our predictions on selectively aggregated context based on paths traversed in the graph by a discrete policy.

Our core insight is that the graph structure of the scene can additionally be leveraged to explicitly model the lateral or route variability in trajectories. We propose a novel approach for trajectory prediction termed Prediction via Graph-based Policy (PGP). Our approach relies on two key ideas.

**Predictions conditioned on traversals:** We selectively aggregate part of the scene context for each prediction, by sampling path traversals from a learned behavior cloning policy as shown in Fig. 1. By more directly selecting the subset of the graph that is used for each prediction, we lessen the representational demands on the output decoder. Additionally, the probabilistic policy leads to a diverse set of sampled paths and captures the lateral variability of the multimodal trajectory distribution.

**Latent variable for longitudinal variability:** To account for longitudinal variability of trajectories, we additionally condition our predictions with a sampled latent variable. This allows our model to predict distinct trajectories even for identical path traversals. We show through our experiments that this translates to greater longitudinal variability of predictions.

We summarize our main contributions on multimodal motion prediction using HD maps:

- A novel method which combines discrete policy roll-outs with a lane-graph subset decoder.
- State-of-the-art performance on the nuScenes motion prediction challenge.
- Extensive ablations demonstrating ability to capture lateral and longitudinal motion variations.

## 2   Related Work

**Graph representation of HD maps:** Most self-driving cars have access to HD vector maps, which include detailed geometric information about objects such as lanes, crosswalks, stop signs, and more. VectorNet [2] encodes the scene context using a hierarchical representation of map objects and agent trajectories. Each component is represented as a sequence of vectors, which are then processed by a local graph network. The resulting features are aggregated globally via a fully-connected graph network. LaneGCN [3] extracts a lane graph from the HD map, and uses a graph convolutional network to compute lane features. These features are combined with both agent and other lane features in a fusion network. Both methods utilize the entire graph for making predictions, relying on the header to identify the most relevant features.

**Multimodal trajectory prediction:** Researchers have proposed a variety of ways to model the multiple possible future trajectories that vehicles may take. One approach is to model the output as a probability distribution over trajectories, using either regression [1], ordinal regression [6], or classification [10]. Another approach models the output as a spatial-temporal occupancy grid [11]. Sampling methods use stochastic policy roll outs [12, 13] or latent variable models that map a latent variable sampled from a simple distribution to a predicted trajectory. Latent variable models are trained as GANs [14, 15], CVAEs [16, 17], or directly using the winner-takes-all regression loss [18]. These models must learn a one-to-many mapping from the entire input context (except the random variable) to multiple trajectories, and can lead to predictions that are not scene compliant.

**Goal-conditioned trajectory prediction:** Rather than learning a one to many mapping from the entire context to multiple future trajectories, methods such as TnT [19], LaneRCNN [20], and PEC-Net [21] condition each prediction on goals of the driver. Conditioning predictions on future goals

makes intuitive sense and helps leverage the HD map by restricting goals to be near the lanes. However, one limitation is that over moderate time horizons, there can be multiple paths that reach a given goal location. Additionally, certain plausible goal locations might be unreachable due to constraints in the scene that are not local to the goal location, e.g., a barrier that blocks a turn lane. In contrast, our method conditions on paths traversed in a lane graph, which ensures that the inferred goal is reachable. Furthermore, the traversed path provides a stronger inductive bias than just the goal location. A similar stream of work conditions on candidate lane centerlines as goals (e.g., WIMP [22], GoalNet [9], CXX [23]). While the lane centerline provides more local context than just the goal, accounting for lane changes can be difficult. Additionally, routes need to be deterministically chosen, with multiple trajectories predicted along the selected route. Our approach allows for probabilistic sampling of both routes and motion profiles. In scenes with just a single plausible route, our model can use its prediction budget of $K$ trajectories purely for different plausible motion profiles. Closest to our work is P2T [24]. They predict trajectories conditioned on paths explored by an IRL policy over a grid defined over the scene. However, they use a rasterized BEV image for the scene, which leads to inefficient encoders and loss of connectivity information due to occlusions. Additionally, their model cannot generate different motion profiles along a sampled path.

## 3 Formulation

We predict the future trajectories of vehicles of interest, conditioned on their past trajectory, the past trajectories of nearby vehicles and pedestrians, and the HD map of the scene. We represent the scene and predict trajectories in the bird's eye view and use an agent-centric frame of reference aligned along the agent's instantaneous direction of motion.

### 3.1 Trajectory representation

We assume access to past trajectories of agents in the scene obtained from on-board detectors and multi-object trackers. We represent the past trajectory of agent $i$ as a sequence of motion state vectors $s^i_{-t_h:0} = [s^i_{-t_h}, ..., s^i_{-1}, s^i_0]$ over the past $t_h$ time steps. Each $s^i_t = [x^i_t, y^i_t, v^i_t, a^i_t, \omega^i_t, \mathcal{I}^i]$, where $x^i_t, y^i_t$ are the BEV location co-ordinates, $v^i_t, a^i_t$ and $\omega^i_t$ are the speed, acceleration and yaw-rate of the agent at time $t$, and $\mathcal{I}^i$ is an indicator with value 1 for pedestrians and 0 for a vehicles. We nominally assign the index 0 to the target vehicle, and timestamp 0 to the time of prediction.

### 3.2 Representing HD maps as lane graphs

**Nodes:** We represent the HD map as a directed graph $\mathcal{G}(V, E)$. The network of lane centerlines captures both, the direction of traffic flow, and the legal routes that each driver can follow. We seek to use both as inductive biases for our model. We thus use lane centerlines as nodes ($V$) in our graph. We consider all lane centerlines within a fixed area around the target vehicle. To ensure that each node represents a lane segment of a similar length, we divide longer lane centerlines into smaller snippets of a fixed length, and discretize them to a set of N poses. Each snippet corresponds to a node in our graph, with a node $v$ represented by a sequence of feature vectors $f^v_{1:N} = [f^v_1, ..., f^v_N]$. Here each $f^v_n = [x^v_n, y^v_n, \theta^v_n, \mathcal{I}^v_n]$, where $x^v_n, y^v_n$ and $\theta^v_n$ are the location and yaw of the $n^{th}$ pose of $v$ and $\mathcal{I}^v_n$ is a 2-D binary vector indicating whether the pose lies on a stop line or crosswalk. Thus, our node features capture both the geometry as well as traffic control elements along lane centerlines.

**Edges:** We constrain edges ($E$) in the lane graph such that any traversed path through the graph corresponds to a legal route that a vehicle can take in the scene. We consider two types of edges. Successor edges ($E_{suc}$) connect nodes to the next node along a lane. A given node can have multiple successors if a lane branches out at an intersection. Similarly, multiple nodes can have the same successor if two or more lanes merge. To account for lane changes, we additionally define proximal edges ($E_{prox}$) between neighboring lane nodes if they are within a distance threshold of each other and their directions of motion are within a yaw threshold. The yaw threshold ensures that proximal edges are not erroneously assigned in intersections where multiple lanes cross each other.

### 3.3 Output representation

To account for multimodality of the distribution of future trajectories, we output a set of $K$ trajectories $[\tau^1_{1:t_f}, \tau^2_{1:t_f}, ..., \tau^K_{1:t_f}]$ for the target vehicle consisting of future x-y locations over a prediction horizon of $t_f$ time steps. Each of the $K$ trajectories represents a mode of the predictive distribution, ideally corresponding to different plausible routes or different motion profiles along the same route.

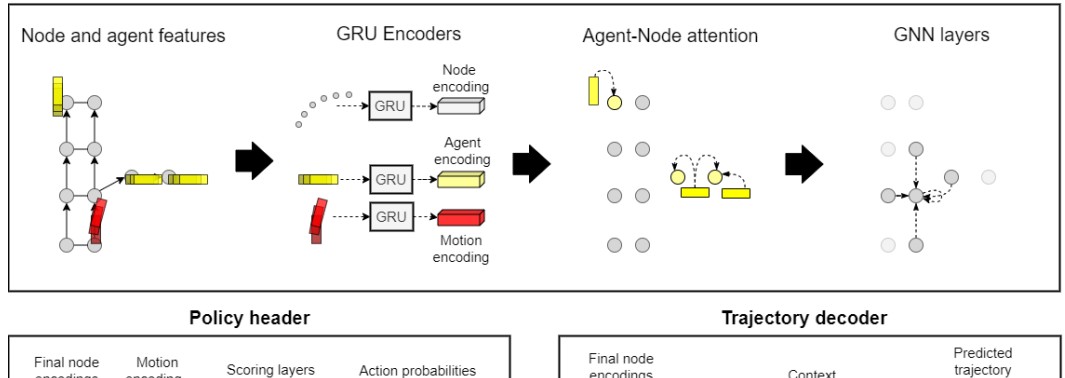

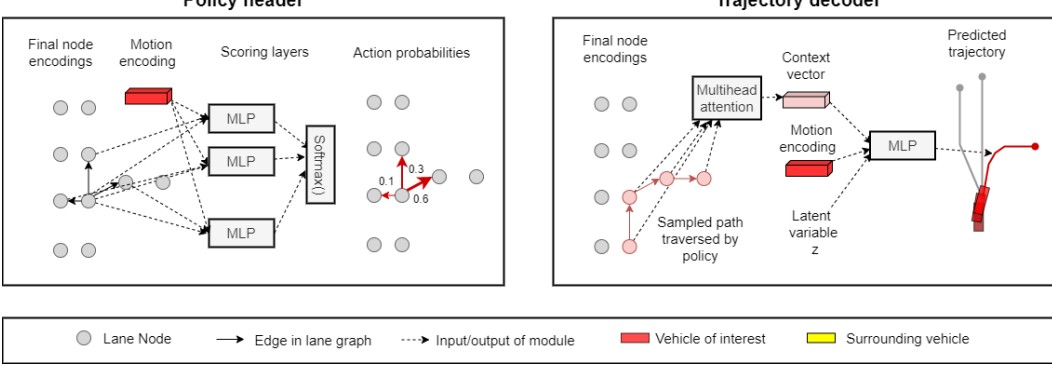

Figure 2: **Proposed model.** PGP consists of three modules trained end-to-end. The graph encoder (top) encodes agent and map context as node encodings of a directed lane-graph. The policy header (bottom-left) learns a discrete policy for sampled graph traversals. The trajectory decoder (bottom-right) predicts trajectories by selectively attending to node encodings along paths traversed by the policy and a sampled latent variable.

## 4 Proposed Model

Fig. 2 provides an overview of our model. It consists of three interacting modules trained end-to-end. The *graph encoder* (Sec. 4.1) forms the backbone of our model. It outputs learned representations for each node of the lane graph, incorporating the HD map as well as surrounding agent context. The *policy header* (Sec. 4.2) outputs a discrete probability distribution over outgoing edges at each node, allowing us to sample paths in the graph. Finally, our attention based *trajectory decoder* (Sec. 4.3) outputs trajectories conditioned on paths traversed by the policy and a sampled latent variable.

### 4.1 Encoding scene and agent context

Inspired by the simplicity and effectiveness of graph based encoders for trajectory prediction [2, 3], we seek to encode all agent features and map features as node encodings of our lane graph $\mathcal{G}(V, E)$.

**GRU encoders.** Both, agent trajectories and lane polylines form sequences of features with a well defined order. We first independently encode both sets of features using gated recurrent unit (GRU) encoders. We use three GRU encoders for encoding the target vehicle trajectory $s^0_{-t_h:0}$, surrounding vehicle trajectories $s^i_{-t_h:0}$ and node features $f^v_{1:N}$. These output the motion encoding $h_{motion}$, agent encodings $h^i_{agent}$ and initial node encodings $h^v_{node}$ respectively.

**Agent-node attention.** Drivers co-operate with other drivers and pedestrians to navigate through traffic scenes. Thus, surrounding agents serve as a useful cue for trajectory prediction. Of particular interest are agents that might interact with the target vehicle's route. We thus update node encodings with nearby agent encodings using scaled dot product attention [4]. We only consider agents within a distance threshold of each lane node to update the node encoding. This allows our trajectory decoder (Sec 4.3) to selectively focus on agents that might interact with specific routes that the target vehicle might take. We obtain keys and values by linearly projecting encodings $h^i_{agent}$ of nearby agents, and the query by linearly projecting $h^v_{node}$. Finally, the updated node encoding is obtained by concatenating the output of the attention layer with the original node encoding.

**GNN layers.** With the node encodings updated with nearby agent features, we exploit the graph structure to aggregate local context from neighboring nodes using graph neural network (GNN) layers. We experiment with graph convolution (GCN) [5] and graph attention (GAT) [25] layers. For the GNN layers, we treat both successor and proximal edges as equivalent and bidirectional. This allows us to aggregate context along all directions around each node. The outputs of the GNN layers serve as the final node encodings learned by the graph encoder.

## 4.2 Discrete policy for graph traversal

Every path in our directed lane graph corresponds to a plausible route for the target vehicle. However, not every route is equally likely. For example, the past motion of the target vehicle approaching an intersection might indicate that the driver is preparing to make a turn rather than go straight. A slow moving lane make it likelier for the target vehicle to change lane rather than maintain lane.

We seek to learn a policy $\pi_{route}$ for graph traversal such that sampled roll-outs of the policy correspond to likely routes that the target vehicle would take in the future. We represent our policy as a discrete probability distribution over outgoing edges at each node. We additionally include edges from every node to an *end* state to allow $\pi_{route}$ to terminate at a goal location. The edge probabilities are output by the policy header shown in Fig. 2. The policy header uses an MLP with shared weights to output a scalar score for each edge $(u, v)$ given by,

$$\text{score}(u, v) = \text{MLP}\left(\text{concat}(h_{motion}, h_{node}^u, h_{node}^v, \mathbb{1}_{(u,v) \in E_{suc}})\right). \tag{1}$$

The scoring function thus takes into account the motion of the target vehicle as well as local scene and agent context at the specific edge. We then normalize the scores using a softmax layer for all outgoing edges at each node to output the policy for graph traversal,

$$\pi_{route}(v|u) = \text{softmax}(\{\text{score}(u, v) | (u, v) \in E\}). \tag{2}$$

We train the policy header using behavior cloning. For each prediction instance, we use the ground truth future trajectory to determine which nodes were visited by the vehicle. We can naively assign each pose in the future trajectory to the closest node in the graph. However, this can lead to erroneous assignment of nodes in intersections, where multiple lanes intersect. We thus only consider lane nodes whose direction of motion is within a yaw threshold of the target agent's pose. An edge $(u, v)$ is treated as visited if both nodes $u$ and $v$ are visited. We use negative log likelihood of the edge probabilities for all edges visited by the ground truth trajectory ($E_{gt}$), as the loss function for training the graph traversal policy, given by

$$\mathcal{L}_{BC} = \sum_{(u,v) \in E_{gt}} -\log(\pi_{route}(v|u)). \tag{3}$$

## 4.3 Decoding trajectories conditioned on traversals

Sampling roll-outs of $\pi_{route}$ yields plausible future routes for the target vehicle. We posit that the most relevant context for predicting future trajectories is along these routes and propose a trajectory decoder that selectively aggregates context along the sampled routes.

Given a sequence of nodes $[v_1, v_2, ..., v_M]$ corresponding to a sampled policy roll-out, our trajectory decoder uses multi-head scaled dot product attention [4] to aggregate map and agent context over the node sequence as shown in Fig. 2. We linearly project the target vehicle's motion encoding to obtain the query, while we linearly project the node features $[h_{node}^{v_1}, h_{node}^{v_2}, ..., h_{node}^{v_M}]$ to obtain keys and values for computing attention. The multi-head attention layer outputs a context vector $\mathcal{C}$ encoding the route. Each distinct policy roll-out yields a distinct context vector, allowing us to predict trajectories along a diverse set of routes.

Diversity in routes alone does not account for the multimodality of future trajectories. Drivers can brake, accelerate and follow different motion profiles along a planned route. To allow the model to output distinct motion profiles, we additionally condition our predictions with a sampled latent vector $z$. Unlike routes, vehicle velocities and accelerations vary on a continuum. We thus sample $z$ from a continuous distribution. We use the multivariate standard normal distribution for simplicity.

Table 1: Comparison to the state of the art on nuScenes

| Model | MinADE$_5$ | MinADE$_{10}$ | MissRate$_{5,2}$ | MissRate$_{10,2}$ | Offroad rate |
|---|---|---|---|---|---|
| CoverNet [10] | 1.96 | 1.48 | 0.67 | - | - |
| Trajectron++ [17] | 1.88 | 1.51 | 0.70 | 0.57 | 0.25 |
| SG-Net [28] | 1.86 | 1.40 | 0.67 | 0.52 | 0.04 |
| MHA-JAM [29] | 1.81 | 1.24 | **0.59** | 0.46 | 0.07 |
| CXX [23] | 1.63 | 1.29 | 0.69 | 0.60 | 0.08 |
| P2T [24] | 1.45 | 1.16 | 0.64 | 0.46 | **0.03** |
| PGP (Ours) | **1.30** | **1.00** | 0.61 | **0.37** | **0.03** |

Finally, to sample a trajectory $\tau_{1:t_f}^k$ from our model, we sample a roll-out of $\pi_{route}$ and obtain $\mathcal{C}_k$, we sample $z_k$ from the latent distribution and concatenate both with $h_{motion}$ and pass them through an MLP to output $\tau_{1:t_f}^k$ the future locations over $t_f$ timesteps,

$$\tau_{1:t_f}^k = \text{MLP}(\text{concat}(h_{motion}, \mathcal{C}_k, z_k)). \tag{4}$$

The sampling process can often be redundant, yielding similar or repeated trajectories. However our light-weight encoder and decoder heads allows us to sample a large number of trajectories in parallel. To obtain a final set of $K$ modes of the trajectory distribution, we use K-means clustering and output the cluster centers as our final set of $K$ predictions $[\tau_{1:t_f}^1, \tau_{1:t_f}^2, ..., \tau_{1:t_f}^K]$. We train our decoder using the winner takes all average displacement error with respect to the ground truth trajectory ($\tau^{gt}$) in order to not penalize the diverse plausible trajectories output by our model,

$$\mathcal{L}_{reg} = \min_k \frac{1}{t_f} \sum_{t=1}^{t_f} \|\tau_t^k - \tau_t^{gt}\|_2. \tag{5}$$

We train our model end-to-end using a multi-task loss combining losses from Eq. 3 and Eq. 5,

$$\mathcal{L} = \mathcal{L}_{BC} + \mathcal{L}_{reg}. \tag{6}$$

## 5   Experiments

**Dataset:** We evaluate our method on nuScenes [26], a self-driving car dataset collected in Boston and Singapore. nuScenes contains 1000 scenes, each 20 seconds, with ground truth annotations and HD maps. Vehicles have manually-annotated 3D bounding boxes, which are published at 2 Hz. The prediction task is to use the past 2 seconds of object history and the map to predict the next 6 seconds. We use the standard split from the nuScenes software kit [27].

**Metrics:** To evaluate our model, we use the standard metrics on the nuScenes leaderboard [27]. The minimum average displacement error (ADE) over the top K predictions (MinADE$_K$). The miss rate (MissRate$_{K,2}$) only penalizes predictions that are further than 2 m from the ground truth. The offroad rate measures the fraction of predictions that are off the road. Since all examples in nuScenes are on the road, this should be zero. Additionally, we report metrics measuring sample diversity of a set of $K$ predictions. To measure lateral diversity, we report the average number of distinct final lanes reached, and the variance of final heading angle of the target vehicle ($\sigma_{yaw}^2$) for the set of $K$ trajectories. To measure longitudinal diversity, we report the variance of average speeds ($\sigma_{speed}^2$) and accelerations ($\sigma_{acc}^2$) for the set of $K$ trajectories.

**Comparison to the state of the art:** We report our results on the standard benchmark split of the nuScenes prediction dataset in table 1, comparing with the top performing entries on the nuScenes leaderboard. We achieve state of the art results on almost all metrics, significantly outperforming the previous best entry P2T [24] on the MinADE$_K$ and MissRate metrics, while achieving comparable off-road rate. This suggests that our model achieves better coverage of the modes of the trajectory distribution, while still predicting trajectories that are scene-compliant.

**Encoder ablations:** We analyze the effects of our graph structure and components of the graph encoder by performing ablations on the graph encoder reported in table 2. In particular we analyze the effect of including proximal edges, modeling surrounding agents with agent-node attention and

Table 2: Encoder ablations

| Graph structure | | Agent-node attention | GNN layers | MinADE$_K$ | | MissRate$_{K,2}$ | | Offroad rate |
|---|---|---|---|---|---|---|---|---|
| $E_{suc}$ | $E_{prox}$ | | | K=5 | K=10 | K=5 | K=10 | |
| ✓ | | | | 1.35 | 1.03 | 0.64 | 0.41 | 0.04 |
| ✓ | ✓ | | | 1.33 | 1.01 | 0.63 | 0.38 | 0.03 |
| ✓ | ✓ | ✓ | | **1.30** | **1.00** | **0.61** | **0.37** | **0.03** |
| ✓ | ✓ | ✓ | GCN × 1 | 1.31 | 1.01 | 0.62 | 0.39 | 0.04 |
| ✓ | ✓ | ✓ | GCN × 2 | 1.31 | 1.01 | 0.61 | 0.39 | 0.04 |
| ✓ | ✓ | ✓ | GAT × 1 | **1.30** | **1.00** | 0.62 | 0.38 | 0.03 |
| ✓ | ✓ | ✓ | GAT × 2 | 1.31 | 1.01 | **0.61** | **0.37** | **0.03** |

Table 3: Decoder ablations

| Decoder | MinADE$_5$ | MinADE$_{10}$ | MissRate$_{5,2}$ | MissRate$_{10,2}$ | Offroad rate |
|---|---|---|---|---|---|
| MTP [1] | 1.59 | 1.12 | **0.57** | 0.48 | 0.08 |
| Latent var (LV) only | 1.38 | 1.08 | 0.65 | 0.43 | 0.05 |
| Traversal only | 1.37 | 1.10 | 0.65 | 0.44 | 0.04 |
| Goals + LV | 1.33 | 1.02 | 0.60 | 0.42 | 0.06 |
| Traversals + LV | **1.31** | **1.01** | 0.61 | **0.37** | **0.03** |

Table 4: Lateral diversity metrics (K=10)

| Decoder | # distinct final lanes | $\sigma^2_{yaw}$ |
|---|---|---|
| LV only | 1.22 | 0.11 |
| Traversals + LV | **1.41** | **0.13** |

Table 5: Longitudinal diversity metrics (K=10)

| Decoder | $\sigma^2_{speed}$ | $\sigma^2_{acc}$ |
|---|---|---|
| Traversal only | 2.33 | 5.28 |
| Traversals + LV | **4.07** | **6.65** |

finally aggregating local context using GCN [5] or GAT [25] layers. We get improvement across all metrics by adding proximal edges, and agent-node attention, suggesting the importance of modeling lane changes and agent context. Somewhat surprisingly, adding GNN layers gives ambiguous results with GCN layers achieving slightly worse results and GAT layers performing on par with the encoder without GNN layers. This could be because the multi-head attention layer aggregates context across the entire traversed path, making the GNNs redundant.

**Decoder ablations:** We next analyze the effect of our traversal and latent variable based decoder. We compare several decoders, all built on top of our proposed encoder with both types of edges, agent-node attention and 2 GAT layers. First, we consider the multimodal regression header from MTP [1]. Next we consider ablations of our decoder without the graph traversals and without the latent variable conditioning. Finally, we consider a model that conditions predictions on sampled goals at different node locations, instead of traversals. Table 3 reports quantitative results while Fig. 3 shows qualitative examples comparing the decoders. We make the following observations.

MTP generally fares worse compared to the other decoders, particularly in terms of offroad rate. We note from Fig. 3 that while it generates a diverse set of trajectories, several veer off-road.

The decoders conditioned purely on the latent variable or purely on traversals both fare worse in terms of MinADE and MissRate compared to our decoder conditioned on both. From the sample diversity metrics (Tables 4 and 5) and qualitative examples (Fig.3) we observe that this is for different reasons. The 'LV only' decoder generates diverse motion profiles, but almost always predicts trajectories along a single route, leading to poor lateral diversity of trajectories. On the other hand, the 'Traversal only' decoder predicts trajectories over a variety of routes, but lacks diversity in terms of motion profiles.

Finally, the 'Goals + LV' decoder also fares worse compared to our 'Traversals + LV' decoder, again, especially in terms of off-road rate. Qualitatively, we observe that this is due to two types of errors. First, it tends to predict spurious goals which aren't reachable for the target vehicle (Fig.3 ③, ④), and second, while it predicts correct goals, it generates trajectories that don't follow accurate paths to those goals (Fig.3 ②,⑥).

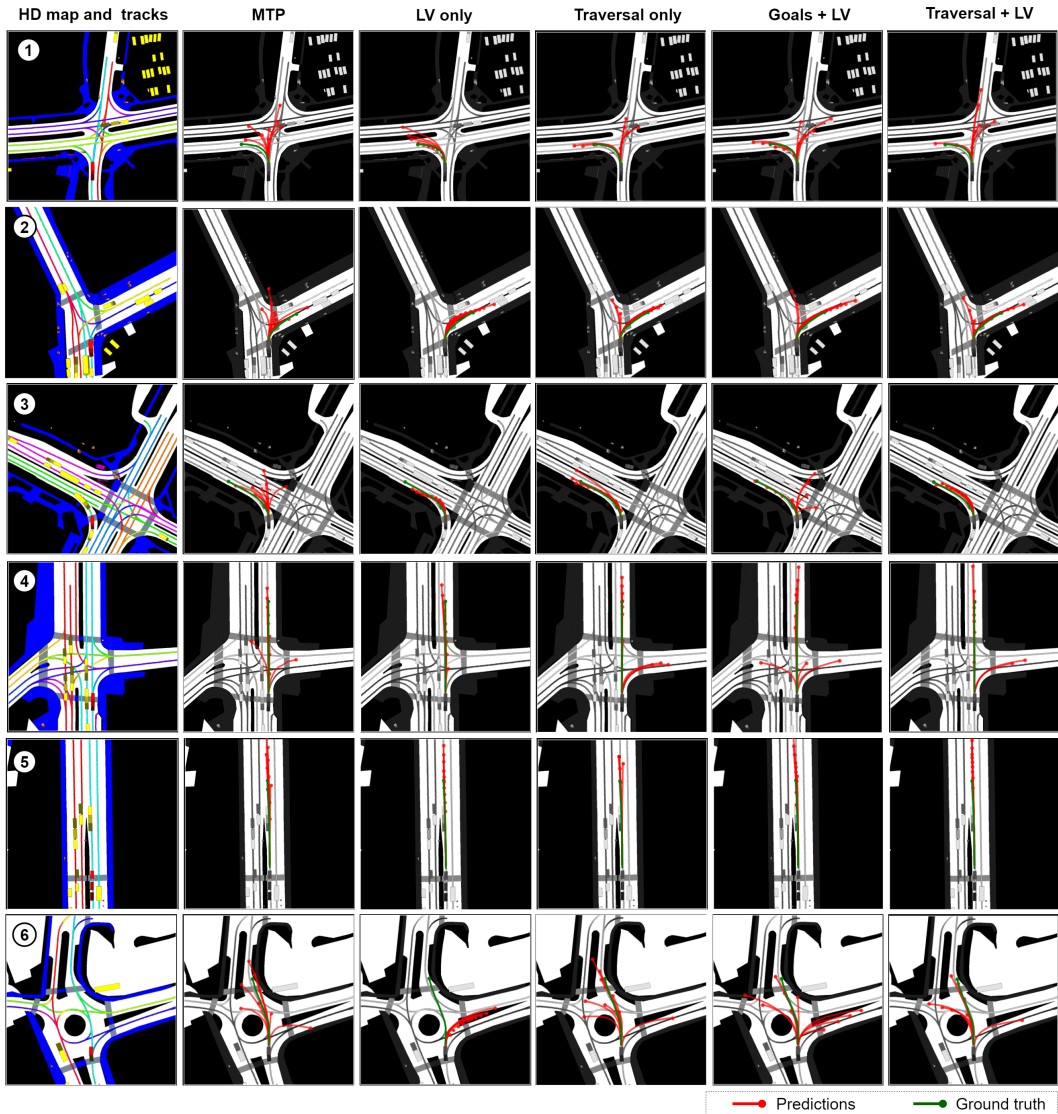

Figure 3: **Qualitative comparison of decoders:** MTP (*column 2*) predicts trajectories that often veer off-road (①-③,⑥). The decoder purely conditioned on latent variables (*column 3*) lacks lateral diversity and predicts trajectories along a single route, even missing the correct route in ⑥. The decoder conditioned purely on traversals (*column 4*) predicts diverse routes, but lacks longitudinal diversity (①,②,⑤). Finally, the decoder conditioned on goals rather than path traversals (*column 5*) predicts spurious goals that may not be reachable (③, ④). Our model (*column 6*) predicts scene-compliant trajectories over a diverse set of routes. In cases with few plausible routes (e.g.⑤), it uses its prediction budget of $K$ trajectories to generate more longitudinal diversity.

## 6  Conclusions

We presented a novel method for multimodal trajectory prediction conditioned on paths traversed in a lane graph of the HD map by a discrete policy, and a sampled latent variable. Through experimental analysis and ablation studies using the publicly available nuScenes dataset, we showed that

- Selectively conditioning predictions on lane-graph traversals leads to trajectories that are (i) diverse in terms of routes, and (ii) precise and scene compliant with the lowest offroad-rates.
- Additionally conditioning predictions on sampled latent variables leads to trajectories that are diverse in terms of motion profiles.
- Both put together lead to state of the art results in terms of MinADE and MissRate metrics.

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
