# OpenReview forum: "Multimodal Trajectory Prediction Conditioned on Lane-Graph Traversals"
_robot-learning.org/CoRL/2021/Conference — CoRL2021 Poster_

### Official Review · Reviewer_qYAi · 2021-07-22

**Originality:** Fair
**Technical Quality:** Fair
**Clarity Of Presentation:** Fair
**Impact:** 2

**Recommendation:**

Weak Reject: I recommend rejecting the paper, but will not argue for my recommendation if the majority of other reviewers have a different opinion.

**Summary:**

This paper focus on the vehicle trajectory prediction, which is one of the necessary parts of autonomous driving technology. In this condition, a future trajectory of a vehicle is predicted based on the environment (e.g., a map of road lane) and past trajectories of other vehicles or pedestrians nearby. While this predicted trajectory is expected to meet the certain constraints such as traffic rules and road topology, it is also involved in uncertainties from other vehicles and pedestrians or drivers’ own behaviors. In this paper, instead of analysing the interaction of these uncertainties with a vehicle, authors decouple them into two parts at the trajectory stage: the lateral changes (e.g., keeping lane, turning) and the longitudinal changes (e.g., accelerating, braking). A model named PGP is therefore raised in this paper, where the lateral variability is captured by a selectively aggregate part with behavior cloning policy based on the features from a graph encoder, and the longitudinal variability is presented by a latent variable model. The prediction on nuScenes dataset shows a SOTA performance.

**Issues:**

1.	The caption of figure 3 is incomplete. Lacking of explanation of figures.
2.	As a precondition of the method proposed in this paper, decoupling uncertainty into lateral and longitudinal (mentioned in line 3-5) is not fully discussed and analysed, or supported by other literature.
3.	One of the two key ideas, predictions conditioned on traversals, is implemented by sampling path traversals from a learned behavior cloning policy (mentioned in line42-47). However, this key idea is insufficiently discussed, only few details is shown in section 4.2 from line 163 to line 178. It could be more convincing if the structure of SVM used in section 4.2 (mentioned in supplementary material), or the formula of loss function used in line 178, and any other necessary details are included in this paper.
4.	One of the two key ideas, latent variable for longitudinal variability, is seemed to be only discussed in section 4.3 from line 190 to line 196. It could be more convincing if the literature of latent variable, or the reason why vector z is sampled from multivariate standard normal distribution, or the comparison with latent variable part in other SOTA models, and any other necessary details are included in this paper.
5.	The multi-agent behavior is a main topic in vehicle trajectory prediction. It is implicitly or explicitly considered in SOTA models (e.g., the model Trajectron++ mentioned in comparison). The model PGP proposed by this paper achieved SOTA results. Therefore, it is expected to indicated how these two key ideas or other features of PGP help to describe and analyse the multi-agent system in trajectory prediction.
6.	In figure 3, as it is described from line 258 to line 260, the model performance seems to be different in a straight road and different types of road junctions. However, this difference is not discussed.

**Reviewer Expertise:**

Good: General knowledge of the area

**Strengths And Weaknesses:**

An encoder-decoder structured model with a policy header included is proposed. The model performs a SOTA results on nuScenes dataset. However, the hypothesis “decoupling uncertainty into lateral and longitudinal” (mentioned in line 3-5) , as the precondition of the method in this paper, has not been discussed by authors. Another main issue is that authors stress two key ideas, predictions conditioned on traversals by behavior cloning and latent variable for longitudinal variability, while these two ideas have not been fully discussed and analysed in the paper, which makes the paper lose focus on its main cocepts.

**Summary Of Recommendation:**

The paper fails to fully discuss its key ideas and show the relationship between the SOTA performance and its ideas. It is recommended to reorganize the paper and focus more on these ideas than other details of medel.

---

> ### Author Response · Authors · 2021-08-28
> **Author response to reviewer qYAi**
>
> Thank you for the comments and suggestions! We have added more discussion on the key concepts in the revised version of the paper. We have addressed specific comments as follows:
>
> **The caption of figure 3 is incomplete. Lacking of explanation of figures.**
>
> We have updated the caption of figure 3 to include more discussion on the contents of the figure, and have also added more annotations and a legend. We hope the figure is self-contained now.
>
> **As a precondition of the method proposed in this paper, decoupling uncertainty into lateral and longitudinal (mentioned in line 3-5)
> is not fully discussed and analysed, or supported by other literature.**
>
> Literature: Decoupling routes (lateral variability) and motion profiles (longitudinal variability) of trajectories is common practice in path planning [7, 8].  More recently, this has also been done for prediction [9], where the authors refer to diverse routes as ‘spatial modes’ and diverse motion profiles as ‘temporal modes’. We have added relevant references to the introduction.
>
> Analysis: We have included additional metrics that measure the lateral (route) and longitudinal (motion) variability of a set of K predicted trajectories. For lateral variability, we report the number of distinct final lanes reached by the set of K trajectories, and the variance of the final heading angle of the target vehicle. For longitudinal variability, we report the variance of average speeds and accelerations of the K trajectories.
>
> We show in Table 4 that a decoder conditioned on traversed paths in the graph achieves greater lateral variability compared to one without it. In Table 5, we show that a decoder conditioned with the sampled latent variable achieves greater longitudinal variability compared to one without it. We hope this provides more evidence for the lateral and longitudinal division of the decoder.
>
> **One of the two key ideas, predictions conditioned on traversals, is implemented by sampling path traversals from a learned behavior cloning policy (mentioned in line 42-47). However, this key idea is insufficiently discussed, only few details is shown in section 4.2 from line 163 to line 178. It could be more convincing if the structure of SVM used in section 4.2 (mentioned in supplementary material), or the formula of loss function used in line 178, and any other necessary details are included in this paper.**
>
> We have added more discussion motivating the discrete policy for sampling routes.
> We have also added the formula for the behavior cloning loss ($\mathcal{L}_{BC}$) along with more discussion on how it is computed.
>
> Also there seems to be some confusion -- we don’t use an SVM for the policy header. The supplementary does not mention an SVM either. Could you please clarify this?
>
> **One of the two key ideas, latent variable for longitudinal variability, is seemed to be only discussed in section 4.3 from line 190 to line 196. It could be more convincing if the literature of latent variable, or the reason why vector z is sampled from multivariate standard normal distribution, or the comparison with latent variable part in other SOTA models, and any other necessary details are included in this paper.**
>
> Sampling based latent variable models trained as GANs [14, 15], CVAEs[16, 17] or normalizing flows [12, 13] have been extensively used for multimodal trajectory prediction. We have added more references in the related works section. We use the latent variable conditioning to allow the model to output diverse motion profiles along the same route. Motion profiles (acceleration, velocities) vary on a continuum, unlike routes which are discrete. We thus use a continuous probability distribution for the latent variable. We use the normal distribution for simplicity, similar to prior work (eg. SocialGAN[14], Desire [16]). We have updated the discussion in section 4.3 to provide more context. Additionally, we *do* compare our model with Trajectron++[17], a latent variable based model representing the SOTA, in table 1.

---

> > ### Author Response · Authors · 2021-08-28
> > **Author response to reviewer qYAi (part 2)**
> >
> > **The multi-agent behavior is a main topic in vehicle trajectory prediction. It is implicitly or explicitly considered in SOTA models (e.g., the model Trajectron++ mentioned in comparison). The model PGP proposed by this paper achieved SOTA results. Therefore, it is expected to indicated how these two key ideas or other features of PGP help to describe and analyse the multi-agent system in trajectory prediction.**
> >
> > We *do* implicitly model the effect of other agents on the target agent. This is the sole purpose of the agent-node attention layers in section 4.1. Furthermore, it leads to better metrics compared to a model without the agent-node attention layers as shown in the encoder ablations (table 2). We have provided more discussion in section 4.1 to clarify this, and motivate it better.
> >
> > **In figure 3, as it is described from line 258 to line 260, the model performance seems to be different in a straight road and different types of road junctions. However, this difference is not discussed.**
> >
> > Our model predicts trajectories along different (reachable) routes at road junctions and intersections. On straight roads with few plausible distinct routes, it uses its prediction budget purely to output different motion profiles. We have added more discussion about this in the caption for figure 3. We hope this addresses the concern.

---

> > > ### Comment · Reviewer_qYAi · 2021-09-02
> > > **comment on paper structure**
> > >
> > > A paper is expected to highlight authors’ ideas. Therefore, I insist that this paper should carefully think about how to improve the structure and the clarity. For example, as authors mentioned in line 43 and line 173, the policy header is trained by using behavior cloning. However, the reason why using behavior cloning is not given. As the main part of this paper, it might be good to carefully discussed and include any necessary explanations.

---

### Official Review · Reviewer_HKDg · 2021-07-22

**Originality:** Very Good
**Technical Quality:** Very Good
**Clarity Of Presentation:** Very Good
**Impact:** 3

**Recommendation:**

Weak Accept: I recommend accepting the paper, but will not argue for my recommendation if the majority of other reviewers have a different opinion.

**Summary:**

The authors propose a method for multimodal trajectory prediction using a graph neural network. They decouple the predicted trajectories components as lateral and longitudinal behaviors and achieve the lateral variability that aggregates part of the scene context by sampling path traversals while ensuring longitudinal diversity via an added latent variable. The experiments are well-presented, and the benchmark evaluations are convincing.

**Issues:**

- Abstract:
The statement "Accurately predicting ... inherent uncertainty in goals and driving behavior. This uncertainty can be loosely decoupled into lateral (e.g., keeping lane, turning) and longitudinal (e.g., accelerating, braking). "It mentions the inherent uncertainty in goals and driving behavior, but the second sentence only explains the uncertainty of driving behaviors.

-Introduction, line 46, "Probabilitic" Is it a typo?

- "Fig. 1.", "Figure 2", "figure 3", "fig. 3" Please be consistent with the notations.

- 4.3 Decoding trajectories conditioned on traversals

The proposed method directly linearly propagates vehicles' motion with latent variable z to achieve longitudinal resilience instead of using the states of the predicted trajectories. But, does it is accurate to qualitatively represent the longitudinal movements?

- Figure 3 is confusing and too small to distinguish the difference. It is recommended to add some notations.

- It is recommended to integrate some details of the supplementary materials into the main content.

- The gif is good to illustrate the information of the proposed method, and it is recommended to add different scenarios with other lateral behaviors.

**Reviewer Expertise:**

Good: General knowledge of the area

**Strengths And Weaknesses:**

The proposed method utilizes the road content as lane graphs. Therefore, the generated trajectories are refined within the road ranges with lower off-road rates. The methodology is concise and with good performance compared with the state-of-the-art on a public dataset and explains the observations of the results. Overall, the paper is sound and well-structured. However, it is unclear about the lateral and longitudinal behavior division on these two decoders, and the implementations about decoder comparison are confusing. There are also some issues to clarify or revise.

**Summary Of Recommendation:**

This paper has done a good job but there are still some weaknesses and issues to solve. But I still recommend this paper to be published.

---

> ### Author Response · Authors · 2021-08-28
> **Author response to reviewer HKDg**
>
> Thank you for the encouraging comments and feedback! We have addressed specific concerns as follows:
>
> **However, it is unclear about the lateral and longitudinal behavior division on these two decoders**
>
> We have included additional metrics that measure the lateral (route) and longitudinal (motion) variability of a set of K predicted trajectories. For lateral variability, we report the number of distinct final lanes reached by the set of K trajectories, and the variance of the final heading angle of the target vehicle. For longitudinal variability, we report the variance of average speeds and accelerations of the K trajectories.
>
> We show in Table 4 that a decoder conditioned on traversed paths in the graph achieves greater lateral variability compared to one without it. In Table 5, we show that a decoder conditioned with the sampled latent variable achieves greater longitudinal variability compared to one without it. We hope this provides more evidence for the lateral and longitudinal division of the decoder.
>
> **Abstract: The statement "Accurately predicting ... inherent uncertainty in goals and driving behavior. This uncertainty can be loosely decoupled into lateral (e.g., keeping lane, turning) and longitudinal (e.g., accelerating, braking)." It mentions the inherent uncertainty in goals and driving behavior, but the second sentence only explains the uncertainty of driving behaviors.**
>
> We consider a prediction horizon of 6 seconds in our experiments. The term ‘goal’ here refers to the goal of the driver over the next 6 seconds. This would typically correspond to the lane that the driver might want to get to (for example turning left, right, changing lane or going straight). However, we understand that this could be confusing without context and have dropped the term ‘goal’ from the abstract.
>
> **4.3 Decoding trajectories conditioned on traversals: The proposed method directly linearly propagates vehicles' motion with latent variable z to achieve longitudinal resilience instead of using the states of the predicted trajectories. But, does it is accurate to qualitatively represent the longitudinal movements?**
>
> Our decoder does not linearly propagate the vehicle’s motion with latent variable z.
> The decoder uses an MLP to output predicted trajectories, conditioned on the vehicle’s past trajectory encoded by a GRU ($h_{motion}$), the latent vector ($z$), and the scene context along routes traversed by the policy ($\mathcal{C}$). The MLP can learn a non-linear mapping from the latent variable to the predicted trajectories. Also, the predictions do take into account the vehicle’s current state via $h_{motion}$.
>
> **Introduction, line 46, "Probabilitic" Is it a typo? "Fig. 1.", "Figure 2", "figure 3", "fig. 3" Please be consistent with the notations.**
>
> Thank you for pointing out the typos! We have fixed them in the revised version.
>
> **Figure 3 is confusing and too small to distinguish the difference. It is recommended to add some notations.**
>
> We have updated figure 3 with more annotations including a legend. We have also updated the figure caption. The caption now provides more discussion on the contents of the figure, making it more self-contained.
>
> **It is recommended to integrate some details of the supplementary materials into the main content.**
>
> We are currently right at the 8 page limit, and it might be difficult to incorporate more details from the appendix to the main text. Was there anything specific from the appendix that you would like added to the main text?
>
> **The gif is good to illustrate the information of the proposed method, and it is recommended to add different scenarios with other lateral behaviors.**
>
> We are happy to note that the gif demonstrating the method was helpful. We have added 7 more gifs to the supplementary material showing predictions over a diverse set of scenes.

---

> > ### Comment · Reviewer_HKDg · 2021-09-02
> > **Response to Author's Comments**
> >
> > I would thank the authors for the helpful response and explanation. But I am still concerned about whether directly linearly propagating vehicles' motion with latent variable z to achieve longitudinal resilience is accurate to qualitatively represent the longitudinal movements. This part should be further explained in this paper. Therefore, I would like to keep my initial recommendation.

---

### Official Review · Reviewer_hmkE · 2021-07-24

**Originality:** Good
**Technical Quality:** Good
**Clarity Of Presentation:** Good
**Impact:** 3

**Recommendation:**

Weak Accept: I recommend accepting the paper, but will not argue for my recommendation if the majority of other reviewers have a different opinion.

**Summary:**

This work focuses on the task of trajectory prediction in a driving context in the presence of multiple road uses. Their approach relies on decomposing the scene into a spatial graph which encodes legal and plausible transitions from one location to another. The key to their approach is that they train a probabilistic prediction policy to generate candidate trajectories in the graph, rather than in coordinate space, and are later transformed to coordinate sequences with another sampling step. By generating probabilistic predictions in graph space and final predictions conditioned on the graph sequences, they more cleanly separate spatial modes.


**Issues:**

I think readers would benefit from a more careful analysis of what makes their approach perform so well. It could be that the success depends less on the novel aspects proposed in the current paper and more on other aspects (i.e. use of transformer, etc...)

**Reviewer Expertise:**

Very good: Comprehensive knowledge of the area

**Strengths And Weaknesses:**

Strengths:
This paper demonstrates state of the art prediction results on the nuScenes benchmark.
They perform an ablation study to demonstrate the effectiveness of various aspects of their approach
Weaknesses:
The paper is selling the premise that their approach of conditioning the final prediction on an encoded graph traversal is the secret sauce. However, in their ablation study they show that conditioning instead on a goal is only slightly worse (1-2cm avg displacement error). If anything their own results show that conditioning the final prediction on a latent variable contributes just as much to their final performance. In fact, all variants in their ablation study outperform the state of the art listed in their baselines.
Would be nice to see performance on other baselines

**Summary Of Recommendation:**

Authors propose novel trajectory prediction approach conditioned on probabilistic graph traversals and achieve state of the art on the nuScenes benchmark. However, from the ablition study it's unclear how much impact the graph traversal truly has. The impact could perhaps be better highlighted by implementing a more diverse set of benchmarks.

---

> ### Author Response · Authors · 2021-08-28
> **Author response to reviewer hmkE**
>
> Thank you for your comments and suggestions! We have addressed the specific comments as follows:
>
> **The paper is selling the premise that their approach of conditioning the final prediction on an encoded graph traversal is the secret sauce. However, in their ablation study they show that conditioning instead on a goal is only slightly worse (1-2cm avg displacement error).**
>
> Compared to the goal conditioned model, our approach achieves lower miss rates and lower off-road rates (last 2 columns of Table 3). The qualitative examples (Fig. 3) show that purely conditioning on goals can lead to trajectories where the goal isn’t reachable, leading to predictions that go off-road or violate traffic rules.
>
> **If anything their own results show that conditioning the final prediction on a latent variable contributes just as much to their final performance. However, from the ablation study it's unclear how much impact the graph traversal truly has**
>
> Conditioning on the latent variable allows the model to produce trajectories with different motion profiles along the same route leading to greater longitudinal variability. Longitudinal variation tends to dominate minADE metrics, which is why conditioning just on the latent variable leads to good minADE values. However as shown in the qualitative examples, the latent variable model’s predictions are consistently unimodal in terms of route. Predictions where the correct route of the target vehicle was missed can be catastrophic for a downstream planner.
>
> We have also added Table 4 to the revised version which reports metrics measuring the lateral diversity of a set of K predictions. Our results show that selectively conditioning on paths traversed by the discrete policy leads to predictions with greater lateral (route) diversity compared to just conditioning predictions with a latent variable
>
> **In fact, all variants in their ablation study outperform the state of the art listed in their baselines. Would be nice to see performance on other baselines. The impact could perhaps be better highlighted by implementing a more diverse set of benchmarks.**
>
> We agree, including an additional benchmark could highlight the impact better. Regrettably, we were not able to add a second benchmark due to time constraints.

---

> > ### Comment · Reviewer_hmkE · 2021-09-05
> >
> > Added information highlights a bit more of the unique contribution of their approach. I stick with my original decision to accept.

---

### Official Review · Reviewer_NjKJ · 2021-07-26

**Originality:** Good
**Technical Quality:** Very Good
**Clarity Of Presentation:** Fair
**Impact:** 4

**Recommendation:**

Weak Accept: I recommend accepting the paper, but will not argue for my recommendation if the majority of other reviewers have a different opinion.

**Summary:**

The paper tackles the problem of agent prediction, an extremely important problem for autonomous driving. The authors leverage the fact that the graph structure of the underlying road scene can be exploited for making trajectory predictions, and they demonstrate how this can be done efficiently by decomposing the trajectory into the lateral and longitudinal components together with clever sampling of candidate trajectories and some other tricks.

**Issues:**

 - Sensitivity analysis with respect to parameter choices
 - Improved clarity of description of method that separates the core generalizable ideas from the heuristics and implementation details.

**Reviewer Expertise:**

Good: General knowledge of the area

**Strengths And Weaknesses:**

Strengths:

[S1] The results seem solid and the ablations are extensive

[S2] From what I can tell, the approach seems novel and the extensions over prior art are well-explained and make intuitive sense

Weaknesses:

[W1] I find some statements hard to understand, or that they lack precision and clarity. E.g.: "Since the entire scene is aggregated together and used to compute all predictions, it places a strong requirement on the output header" what requirement?

[W2] There seem to be a lot of parameters and hyperparameters in the approach. E.g.
 - "We consider all lane centerlines within an area of [-50, 50] m laterally and [-20, 80] m longitudinally around the vehicle of interest. We divide the lane centerlines into snippets of length ≤ 20m, and discretize them to a set of N poses with 1m resolution"
 - "For each node v, we only consider nearby agents within 10m of the lane centerline"
 - "with direction of motion within a yaw threshold of the pose"
The prevalence of all of these heuristics calls into question the robustness of the proposed approach. For each of these choices it would strengthen the work to have some more context or analysis of the sensitivity of the method to the parameters.

[W3] The method seems quite complicated and the architecture used is quite complex with many learned compenents. It is fine that it takes a complex solution to solve a complex problem, but I feel the manuscript could do a better job of elevating the discussion so that we may learn some more generally applicable lessons rather than the step by step recipe for solving the exact problem that the authors set out to solve.


[W4] Minor issues:
probabilitic -> probabilistic
"encodes the scene context a hierarchical representation of" a word is missing somewhere
"fig 3" vs. "figure 3" be consistent.
"the model tend" -> the model tends


**Summary Of Recommendation:**

In summary, I think that the approach seems novel and the results demonstrate its effectiveness. However, I think that the work could be elevated and that the clarity of the exposition could be improved. At present, it comes across as clever set of heuristics and a particularly good job of engineering the components to work together.

---

> ### Author Response · Authors · 2021-08-28
> **Author response to reviewer NjKJ**
>
> Thank you for your comments and suggestions! We have updated the paper to improve the clarity of presentation, with a greater focus on generalizable ideas rather than implementation details. We have addressed the specific comments as follows:
>
>
> **"Since the entire scene is aggregated together and used to compute all predictions, it places a strong requirement on the output header" what requirement?**
>
> Our choice of word “requirement” here was unclear and we have revised the text. Thanks for pointing this out. Here is what we meant to say:
>
> Prior work such as LaneGCN[3] and VectorNet[2] aggregates the entire scene context into a single context vector using either graph convolution (LaneGCN) or attention (VectorNet). An output header like MTP[1] then needs to learn a mapping from this context vector representing the entire scene, to multiple plausible future trajectories. This is a complex mapping and can often lead to several off-road predictions (see table 1 row 1 off-road rate, and figure 3 column 2). We have updated the text in the introduction to clarify this.
>
>
> **There seem to be a lot of parameters and hyperparameters in the approach. For each of these choices it would strengthen the work to have some more context or analysis of the sensitivity of the method to the parameters.**
>
> We have moved specific details and heuristics to the appendix in the supplementary. We have also provided more context justifying the hyperparameter choices detailed below. We will additionally include hyperparameter sweeps for each of these parameters in the appendix. These would take longer than rebuttal time-window. We will do our best to include the analysis for the camera ready version.
>
>
> **“We consider all lane centerlines within an area of [-50, 50] m laterally and [-20, 80] m longitudinally around the vehicle of interest”**
>
> The map extent around the target agent was chosen to ensure that most ground truth future trajectories in the nuScenes train set lie within the selected extent. The specific extent has been moved to appendix A.1, along with the explanation.
>
>
> **“We divide the lane centerlines into snippets of length ≤ 20m, and discretize them to a set of N poses with 1m resolution"**
>
> There is a trade-off associated with the resolution of lane nodes. A finer resolution would provide a more informative set of inputs, but would lead to a graph with a greater number of nodes (and a greater number of poses per node) increasing encoder complexity. The specific resolution values have been moved to appendix A.1, along with the explanation.
>
>
> **"For each node v, we only consider nearby agents within 10m of the lane centerline"**
>
> Our thought process here, similar to LaneGCN[3], was to only incorporate local agent context into each lane node. More global context is aggregated by the decoder over sampled path traversals. This allows the decoder to selectively focus on only those agents that might interact with routes sampled by the policy.  We have updated the text in section 4.1 to clarify this.
>
>
> **“With direction of motion within a yaw threshold of the pose"**
>
> For each prediction instance, we use the ground truth future trajectory to determine which nodes were visited by the vehicle. We can naively assign each pose in the future trajectory to the closest node in the graph. However, this can lead to erroneous assignment of nodes in intersections, where multiple lanes intersect.  We thus only consider lane nodes whose direction of motion is within a yaw threshold of pi/4 of the target agent’s pose. This explanation has been added to sections 4.2 (and in section 3.2 since this also applies to proximal edges).
>
>
> **Minor issues: probabilitic -> probabilistic "encodes the scene context a hierarchical representation of" a word is missing somewhere "fig 3" vs. "figure 3" be consistent. "the model tend" -> the model tends**
>
> Thank you for pointing out the typos! We have fixed them in the revised version.
>
>
> **I feel the manuscript could do a better job of elevating the discussion so that we may learn some more generally applicable lessons rather than the step by step recipe for solving, the exact problem that the authors set out to solve.**
>
> Thank you for this important comment!
>
> There are two key take-aways or generally applicable lessons:
> 1. Selectively conditioning predictions on lane-graph traversals leads to trajectories that are (i) diverse in terms of routes, and (ii) precise and scene compliant with the lowest offroad-rates.
> 2. Additionally conditioning predictions on sampled latent variables leads to trajectories that are diverse in terms of motion profiles.
>
> We have provided further analysis reporting sample diversity metrics measuring lateral and longitudinal diversity of a set of K predicted trajectories in tables 4 and 5. We have also provided more motivation and context for our design choices in sections 3 and 4. Finally we have also itemized the key take-aways in the conclusions section of the paper.

---

> > ### Comment · Reviewer_NjKJ · 2021-09-02
> > **Response to the response**
> >
> > I thank the authors for their detailed replies and for considering my suggestions. Overall I think the paper has improved but the weaknesses related to hyperparameter sensitivity and other issues raised by other reviewers remain. Therefore, I would maintain my previous ranking.

---

### Meta-Review · Area_Chair_haPW · 2021-08-14

**Recommendation:** Accept (Poster)
**Confidence:** 4

**Metareview:**

This paper proposes a novel method for multimodal trajectory prediction. While the reviewers agree that this paper contains some interesting ideas and the proposed method is effective, they also raise a number of important concerns. The authors should carefully address the reviewers' comments in their rebuttal.

UPDATE POST DISCUSSION PHASE: I would like to thank the authors for their comments and clarifications during the discussion phase. The reviewers generally agree that this paper provides a valuable contribution, and I concur with this assessment. In the final version of this paper, the authors should carefully address all reviewers' comments and suggestions, in particular regarding hyperparameter sensitivity and exposition clarity.

---

> ### Author Response · Authors · 2021-08-28
> **Author response to the area chair and reviewers**
>
> We thank the area chair and reviewers for their valuable comments and feedback! We are encouraged that the reviewers see the novelty of the approach (NjKJ), find it concise (HKDg), intuitive (NjKJ) and well-explained (HKDg, NjKJ). We are also happy to note that reviewers have listed the SOTA results on nuScenes (HKDg, hmKE, NjKj, qYAi) and ablations (NjKJ, hmKE, HKDg) as strengths of the paper. The reviewers raised important concerns asking for more discussion of the key ideas of the paper and generally applicable lessons, and more context/motivation for some design choices. We have updated the paper to provide more relevant references in the introduction and related work sections (Sec.1, 2), more motivation for design choices and hyperparameters in the method sections (Sec. 3, 4) and more discussion as well as additional analysis (Tables 4, 5) in the experiments section (Sec. 5). More specific comments are included in the responses to each reviewer below.

---

### Decision · Program_Chairs · 2021-09-13

**Decision:**

Accept (Poster)

**Comment:**

This paper proposes a novel method for multimodal trajectory prediction. While the reviewers agree that this paper contains some interesting ideas and the proposed method is effective, they also raise a number of important concerns. The authors should carefully address the reviewers' comments in their rebuttal.

UPDATE POST DISCUSSION PHASE: I would like to thank the authors for their comments and clarifications during the discussion phase. The reviewers generally agree that this paper provides a valuable contribution, and I concur with this assessment. In the final version of this paper, the authors should carefully address all reviewers' comments and suggestions, in particular regarding hyperparameter sensitivity and exposition clarity.